# SAFER: Data-Efficient and Safe Reinforcement Learning Through Skill Acquisition

## Abstract

Though many reinforcement learning (RL) problems involve learning policies in settings that are difficult to specify safety constraints and sparse rewards, current methods struggle to rapidly and safely acquire successful policies. *Behavioral priors*, which extract useful policy primitives for learning from offline datasets, have recently shown considerable promise at accelerating RL in more complex problems. However, we discover that current behavioral priors may not be well-equipped for safe policy learning, and in some settings, may promote *unsafe* behavior, due to their tendency to ignore data from undesirable behaviors. To overcome these issues, we propose *SAFEty skill pRiors (SAFER)*, a behavioral prior learning algorithm that accelerates policy learning on complex control tasks, under safety constraints. Through principled contrastive training on safe and unsafe data, SAFER learns to extract a *safety variable* from offline data that encodes safety requirements, as well as the *safe primitive skills* over abstract actions in different scenarios. In the inference stage, SAFER composes a safe and successful policy from the safety skills according to the inferred safety variable and abstract action. We demonstrate its effectiveness on several complex safety-critical robotic grasping tasks inspired by the game Operation,[1] in which SAFER outperforms baseline methods in learning successful policies and enforcing safety.

## 1 Introduction

Reinforcement learning (RL) has demonstrated impressive performance at solving complex control tasks. However, RL algorithms still require large amounts of exploration and data collection in order to acquire successful policies. For many complex safety-critical applications (i.e, autonomous driving, healthcare, factory robotics) extensive interactions with an environment are not possible due to potential dangers associated with exploration or the material costs of online data collection. These challenges are further complicated by the fact that it is difficult to manually specify safety constraints in these, due to the complexities of the environments. Nevertheless, relatively few existing safe reinforcement learning algorithms can *rapidly* and *safely* solve challenging, high-dimensional RL problems, with difficult to specify safety constraints.

One promising route is the *behavioral prior* (Singh et al., 2021) for offline skill discovery (Pertsch et al., 2020; 2021; Ajay et al., 2021). These methods use offline datasets to learn representations of useful actions or *behaviors* through generative models, such as a normalizing flow model or variational autoencoders (VAE). Specifically, they treat the latent space of the generative model as the abstract action space of higher level actions (i.e., skills). Equipped with a behavioral prior, for each downstream task, an RL agent is trained to map states onto the abstract action space of skills. This approach can greatly accelerate policy learning because it first learns useful primitives from a dataset. This structure simplifies the action space for RL (Dulac-Arnold et al., 2015).

Intuitively, if trained on datasets consisting of trajectories which are both safe and successful, behavioral priors should capture *safe and useful* behaviors and thus encourage the *rapid* acquisition of safe policies when used to learn future tasks (*downstream* learning). For example, when trained on demonstrations of common house hold tasks, behavioral priors should capture behaviors that successfully and safely accomplish similar tasks, such as handling objects carefully and avoiding animals/people in the environment. However, we find many existing behavioral priors may not be

---

[1] https://en.wikipedia.org/wiki/OperationGame

well suited for safe policy learning. This is mainly because current state-of-the-art behavioral priors (Singh et al., 2021) cannot distinguish between safe and unsafe actions and are unable to avoid the concentration of unsafe actions in high likelihood regions of the abstract action space (see Figure 1). When the behavioral priors (oftentimes modeled with deep generative models) are trained only with safe experiences, the unsafe data is out of the training distribution. It is well known that deep generative models have problems generalizing to out of distribution data, which indeed increases the likelihood of unsafe actions in this case (Nalisnick et al., 2018; Fetaya et al., 2020; Kirichenko et al., 2020). For the household robotics example, the behavioral prior may lead to unsafe behavior (breaking objects or harming animals/people).

In this work, we propose *SAFER*: safety skill priors, which possesses both desiderata: *accelerating* reinforcement learning with *safe* operations. (An overview of SAFER is provided in Figure 2.) To acquire safe behavioral priors, SAFER **i)** uses a contrasive loss to distinguish safe date from unsafe ones and **ii)** learns a posterior sampling distribution of a latent *safety variable*, that captures different safety contexts. With this in hand, SAFER maps the abstract action space onto the set of safe behavioral actions. These safe actions are parameterized by the safety context variable, which makes the behavioral prior more adaptable to different tasks and safety constraints. To further establish the safety assurance of SAFER, we propose a technique to adjust the abstract action space, such that at any state $s$, at most $(1 - \epsilon)\%$ of actions generated by the safety prior is unsafe. As shown in Figure 1, SAFER assigns much lower likelihood to unsafe states and actions,

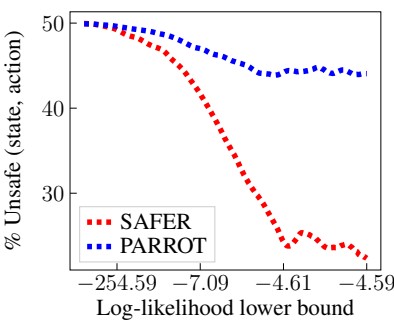

Figure 1: **Evaluating the concentration of unsafe data in high likelihood regions** by computing the % of unsafe state-action pairs in a holdout dataset of a safe robotic grasping task. PARROT assigns high likelihoods to unsafe data, i.e., it does not encourage safety, while SAFER has much lower likelihood in unsafe data.

indicating that it will better promote safe behaviors when applied to downstream RL. For household robotics, this means SAFER is much better equipped to learn behaviors that are both safe and useful for downstream tasks (handling objects safely and not harming animals/people) compared to the alternatives because it may be difficult to manually specify a constraint function, but could easier to access demonstrations of unsafe behavior. To demonstrate the effectiveness of SAFER, we evaluate it on a set of complex safety-critical robotic grasping tasks. When compared with other baseline methods, SAFER policies have a higher success rate and less safety violations.

## 2 BACKGROUND

In a setting with different tasks, for each task $\mathcal{T}$, the agent's interaction is modeled as a Markov decision process (MDP). A MDP is a tuple $(\mathcal{S}, \mathcal{A}, \mathrm{T}, r, \gamma, \boldsymbol{s}_0)$. In the MDP, $\mathcal{S}$ and $\mathcal{A}$ are the state and action spaces, $\mathrm{T}(\cdot | \boldsymbol{s}, \boldsymbol{a})$ is the transition probabilities, $r(\boldsymbol{s}, \boldsymbol{a})$ is the reward function, $\gamma \in [0, 1)$ is the discount factor, and $\boldsymbol{s}_0 \in \mathcal{S}$ is the initial state. A number of different safe RL formulations exist in the literature (García et al., 2015). In this work, safety is measured by the *safety violation function*, $\omega(\boldsymbol{s}, \boldsymbol{a}) \in \{0, 1\}$, that indicates whether the current state and action lead to a safety violation (1) or no safety violation (0). Consequently, the safety MDP is given as a tuple $(\mathcal{S}, \mathcal{A}, \mathrm{T}, r, \gamma, \boldsymbol{s}_0, \omega)$. Note, that the safety violation function $\omega(\boldsymbol{s}, \boldsymbol{a})$ is task specific. This formulation is desirable because different tasks likely have different safety criterion. For instance, in robotic grasping, different tasks likely contain different objects that have various sensitivities to being gripped or limits to how quickly they can move without breaking. Observe that we utilize a binary notion of safety violation because we are concerned with complex RL problems, (e.g., learning from pixels), where it can be difficult to specify continuous notions of safety.

We now formalize the safety MDP problem for each task in the environment. Let $\Delta$ be the set of Markovian stationary policies, i.e., $\Delta = \{\mu : \mathcal{S} \times \mathcal{A} \to [0, 1], \sum_{\boldsymbol{a}} \mu(\boldsymbol{a} | \boldsymbol{s}) = 1\}$. Given a policy $\mu \in \Delta$, we define the expected return as $\mathcal{R}_\mu(\boldsymbol{s}_0) := \mathbb{E}[\sum_{t=0}^{\infty} \gamma^t r(\boldsymbol{s}_t, \boldsymbol{a}_t) \mid \mu, \boldsymbol{s}_0]$ and at each given state $\boldsymbol{s} \in \mathcal{S}$ the safety constraint function (i.e., expected safety violation) as $\mathcal{W}_\mu(\boldsymbol{s}) := \mathbb{E}[\omega(\boldsymbol{s}, \boldsymbol{a}) \mid \mu, \boldsymbol{s}]$. The *safety constraint* is then defined as $\mathcal{W}_\mu(s) \leq \epsilon$, where $\epsilon \in [0, 1]$ is the tolerable threshold of

violation. For each task the goal in safety MDP is to solve the constrained optimization problem

$$\mu^* \in \arg\max_{\mu \in \Delta} \{\mathcal{R}_\mu(\boldsymbol{s}_0) : \mathcal{W}_\mu(\boldsymbol{s}) \leq \epsilon, \; \forall \boldsymbol{s} \in \mathcal{S}\}. \tag{1}$$

It has been shown that if the feasibility set is non-empty, then there exists an optimal policy in the class of stationary Markovian policies $\Delta$ (Altman, 1999, Theorem 3.1). Similar to policy gradient (PG) algorithms, to effectively solve for the optimal policy we parameterize the stationary Markovian policy by a $\kappa$-dimensional vector $\psi$, so the space of policies can be written as $\{\mu_\psi, \; \psi \in \Psi \subset \mathbb{R}^\kappa\}$. In the next section we will further exploit the connections of different tasks to design a parameterization structure with policy primitives that leads to effective and safe learning.

**Behavioral Priors** To effectively solve the safety MDP for each task, similar to the recent work of *primitive discovery* (Singh et al., 2021; Ajay et al., 2021), we propose the specific policy structure $\mu_\psi = f_\phi(\boldsymbol{z}; \boldsymbol{s})$ and $\boldsymbol{z} \sim \pi_\theta(\boldsymbol{z}|\boldsymbol{s})$, where $\psi = (\phi, \theta)$. In this parameterization, the mapping (with learnable parameters $\phi$)

$$f_\phi : \mathcal{Z} \times \mathcal{S} \to \mathcal{A} \tag{2}$$

is denoted as the *behavioral prior* (Singh et al., 2021), which maps from the abstract action space $\mathcal{Z}$ and state space $\mathcal{S}$ to the action space $\mathcal{A}$. The task-dependent, high-level policy $\pi_\theta : \mathcal{S} \to \mathbb{P}(\mathcal{Z})$ maps any state $\boldsymbol{s} \in \mathcal{S}$ to the corresponding distribution of abstract actions in $\mathcal{Z}$. Notice that the behavioral prior $f_\phi$ is independent of any tasks. To train this action mapping more effectively, one can use an offline dataset $\mathcal{D}$, which consists of state-action rollouts $\tau = \{\boldsymbol{s}_0, \boldsymbol{a}_0, ..., \boldsymbol{s}_t, \boldsymbol{a}_t\}$ collected across different tasks. As long as these rollouts are generated by following a diverse set of policies, they will likely contain information for learning different low-level action mappings from $\mathcal{Z} \times \mathcal{S}$ to $\mathcal{A}$ that leads to useful behavior for the downstream tasks. Typically, a simple distribution is chosen as $\mathcal{Z}$ (e.g., the unit normal distribution) to make controlling the primitives straightforward.

Policies trained with behavioral priors use $\mathcal{Z}$ as the action space and optimize the parameters $\theta$ in the policy $\pi_\theta$. In the context of the safety MDP, the behavioral prior provides a simplified action space $\mathcal{Z}$ which only contains actions in the action space $\mathcal{A}$ that will satisfy the safety threshold $\mathcal{W}_\mu(s) := \mathbb{E}[\omega(\boldsymbol{s}, \boldsymbol{a}) \mid \mu, \boldsymbol{s}]$ and maximize rewards $\mathcal{R}_\mu(\boldsymbol{s}_0)$. For example, in a robotic grasping task, a behavioral prior may learn to reach toward objects or grip nearby objects. These properties make behavioral priors highly appealing for solving complex control tasks because learning successful policies from high level behaviors typically requires much less interaction with the environment than learning policies from scratch (Singh et al., 2021). As such, different ways to express the behavior prior mapping have been proposed. For instance, Ajay et al. (2021) optimizes the likelihood of actions, conditioned on the state and abstract action space, $\log \pi_\theta(\boldsymbol{a}|\boldsymbol{s}, \boldsymbol{z})$. Singh et al. (2021) directly optimizes the conditional log-likelihood policy, $\log p(\boldsymbol{a}|\boldsymbol{s})$, and fix an invertible mapping through the use of a conditional normalizing flow (Dinh et al., 2017) between the abstract action space $\mathcal{Z}$ and the distribution over useful actions $p(\boldsymbol{a}|\boldsymbol{s})$ to retain control of the learned behaviors.

**Shortcomings of Behavioral Priors for Safe RL** Though current behavioral priors are highly useful at accelerating learning, they only *increase* of likelihood of useful actions. Thus, when applied to a safety MDP problem, data containing unsafe or unsuccessful data should not be used because it is counter-intuitive to increase the likelihood of these actions (Singh et al., 2021). Consequently, unsafe states and actions may be out of distribution (OOD). It is well established in the literature on deep generative models (including the behavioral prior models) that OOD data is handled poorly and, in some cases, might have higher likelihood than in-distribution data (Nalisnick et al., 2018; Fetaya et al., 2020; Kirichenko et al., 2020). As we see in Figure 1, these observations hold true for current behavioral priors where unsafe data has high likelihood, indicating that they may encourage unsafe behavior in the presence of OOD states. Since the proposed policy structure with behavioral priors relies on high likelihood actions from the prior (Ajay et al., 2021; Singh et al., 2021), using the aforementioned behavior priors to solve safety MDP problems will generally be problematic.

## 3  SAFER: SAFETY SKILL PRIORS

Considering the shortcomings mentioned in Section 2 of existing behavioral priors and the need for methods that can learn complex safety constraints, ideally a behavioral prior that encourages safety should **i)** be capable of learning complex safety constraints by sufficiently exploiting the data, thereby avoiding the OOD issue; **ii)** permit the specification undesirable behaviors through data; and **iii)** accelerate the learning of successful policies. Motivated by these requirements, in this section,

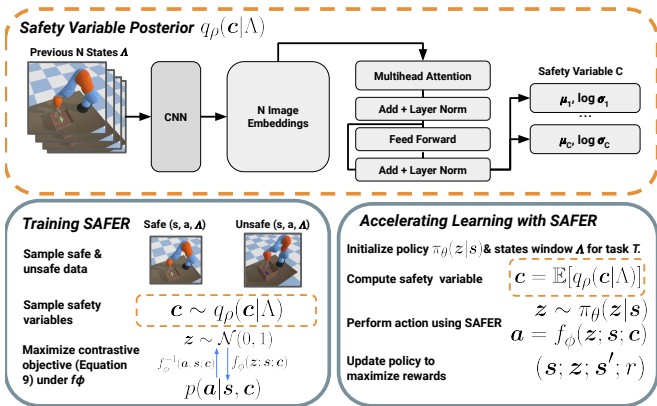

Figure 2: **Overview of SAFER:** SAFER optimizes the posterior over a latent *safety variable* (top of figure) that encodes safety information about the environment. SAFER uses the safety variable to learn an abstract action space $\mathcal{Z}$ that maps to safe and useful behaviors through $f_\phi$ through a normalizing flow (lower left). SAFER accelerates policy learning on downstream tasks by learning to take actions in $\mathcal{Z}$ using policy $\pi_\theta(z|s)$ (lower right).

we introduce SAFER, a behavioral prior that circumvents the aforementioned shortcomings and is specifically designed for safety MDPs.

## 3.1 LATENT SAFETY VARIABLE

To address all of the above criteria, we introduce an additional latent variable as the input to the behavioral prior called the *safety variable* $c \in \mathcal{C}$, i.e.,

$$f_\phi : \mathcal{Z} \times \mathcal{C} \times \mathcal{S} \to \mathcal{A}.$$

The safety variable captures information for the behavioral prior so that it can safely and rapidly generalize when used in downstream tasks. This construction is useful because information beyond the current state $s$ can be encoded into the task variable, to better help the behavioral prior model complex per task safety dynamics. For example, the safety variable could encode the locations of other vehicles in autonomous driving or animals/people in a robotics application. Because we do not assume the task variable $\mathcal{C}$ is provided from the environment, we train a network to infer it.

## 3.2 LEARNING THE SAFETY VARIABLE

In order to train the behavioral prior and posterior over the safety variable, we adopt a variational inference (VI) approach. We jointly train an invertible conditional normalizing flow $f_\phi$ (Dinh et al., 2017) as the behavior prior and posterior over the safety variable using VI. At each state $s \in \mathcal{S}$ and safety variable $c \in \mathcal{C}$, the flow model $f_\phi$ maps a unit Normal abstract action $z \in \mathcal{Z}$ (i.e., samples $z = f_\phi^{-1}(a|s,c)$ of the inverse flow model follow the distribution $p_{\mathcal{Z}}(\cdot) := \mathcal{N}(0, I)$) onto the action space $\mathcal{A}$ of safe behaviors, and therefore the corresponding prior action distribution is given by

$$p_\phi(a|s,c) := p_{\mathcal{Z}}(f_\phi^{-1}(a;s;c)) \cdot |\det(\partial f_\phi^{-1}(a;s;c)/\partial a)|. \tag{3}$$

The flow model is a good choice for the behavioral prior because it allows computing exact log likelihoods. Further, it yields a mapping such that actions taken in the abstract action space $z \in \mathcal{Z}$ can easily be transformed into useful ones $a = f_\phi(z;s;c)$. However, since VI approximates the lower bound of maximum likelihood, it does not explicitly enforce the safety requirements in the safety variable $c$. To overcome this issue, we encode safety to $c$ by formulating the learning problem as chance constrained optimization (Charnes and Cooper, 1959).

**Chance Constrained Objective** Formally, our objective arises from optimizing a neural network to infer the posterior over the safety variable $\mathcal{C}$ using amortized variational inference (Zhang et al., 2019). In particular, we parameterize the posterior over the safety variable as $q_\rho(c|\Lambda)$, where $c$ is the safety variable, and $\Lambda$ is information from which to infer the variable. In practice, we set $\Lambda$ as a sliding window of states, such that if $s_t$ is the current state at time $t$ and $w$ is the window size, then the information is given by $\Lambda = [s_t, s_{t-1}, ..., s_{t-w}]$. We infer the safety variable from the sliding window of states $\Lambda$ because we expect this to contain useful information concerning safe learning. For example, in a robotics setting where the observations are images, previous states may contain useful information concerning the locations of objects to avoid, which may be unobserved in the current state. We write the evidence-lower bound (ELBO) of our model as follows,

$$\mathbb{E}_{c \sim q_\rho(c|\Lambda)}[\log p_\phi(a|s,c) - D_{\mathrm{KL}}(q_\rho(c|\Lambda)||p(c))] \tag{4}$$

where $a$ is the action that will not result in safety violations (i.e, $\omega(a, s) = 0$) and $p(c)$ is a prior over the safety variable $c$. To ensure that unsafe actions are low probability, we add a chance constraint

about the likelihood of unsafe actions (Nemirovski and Shapiro, 2007) to the ELBO optimization,

$$\max_{\rho,\phi} \mathbb{E}_{\boldsymbol{c}\sim q_\rho(\boldsymbol{c}|\boldsymbol{s})}\left[\log p_\phi(\boldsymbol{a}|\boldsymbol{s},\boldsymbol{c}) - D_{\mathrm{KL}}\left(q_\rho(\boldsymbol{c}|\Lambda)||p(\boldsymbol{c})\right)\right] \text{ s.t. } \mathbb{P}_{\boldsymbol{c}}(p_\phi(\boldsymbol{a}_{\mathrm{unsafe}}|\boldsymbol{s},\boldsymbol{c}) > \epsilon) \le \xi, \quad (5)$$

where the constraint states that in most cases (at least with probability $\xi$) with the safety variable $c$ drawn from $\mathcal{C}$ the distribution of the corresponding unsafe actions (i.e., $\omega(\boldsymbol{a}_{\mathrm{unsafe}}, \boldsymbol{s}) = 1$) is always less than the safety threshold $\epsilon$. Intuitively, this objective enforces that the safety variable should make safe actions as likely as possible while minimizing the probability of unsafe actions.

**Tractable Lower Bound**  Due to the difficulty in optimizing the chance constrained ELBO objective function in high-dimensional settings, we instead consider optimizing an unconstrained surrogate lower bound (Nemirovski and Shapiro, 2007).

**Proposition 3.1** *Assuming the chance constrained ELBO is written as in Equation 5, we can write the surrogate lower bound as,*

$$\max_{\rho,\phi,\lambda} \mathbb{E}_{\boldsymbol{c}\sim q_\rho(\boldsymbol{c}|\boldsymbol{s})}\left[\log p_\phi(\boldsymbol{a}|\boldsymbol{s},\boldsymbol{c}) - \lambda \log p_\theta(\boldsymbol{a}_{unsafe}|\boldsymbol{s},\boldsymbol{c}) - D_{KL}(q_\rho(\boldsymbol{c}|\Lambda)||p(\boldsymbol{c}))\right] \qquad (6)$$

**Proof:** We rewrite the optimization 5 into the following form (with the term $+\lambda'\xi$ dropped because it is independent of the decision variables),

$$\max_{\rho,\phi,\lambda'} \mathbb{E}_{\boldsymbol{c}\sim q_\rho(\boldsymbol{c}|\boldsymbol{s})}\left[\log p_\phi(\boldsymbol{a}|\boldsymbol{s},\boldsymbol{c}) - D_{\mathrm{KL}}\left(q_\rho(\boldsymbol{c}|\Lambda)||p(\boldsymbol{c})\right)\right] - \lambda'\mathbb{P}_{\boldsymbol{c}}(p_\phi(\boldsymbol{a}_{\mathrm{unsafe}}|\boldsymbol{s},\boldsymbol{c}) > \epsilon). \quad (7)$$

where $\lambda' > 0$ is a $\xi$-dependent penalty parameter that is connected to the Langrange multiplier[2] and is chosen to enforce the constraint. Using the Markov inequality we have

$$\mathbb{P}_{\boldsymbol{c}}(\ p_\phi(\boldsymbol{a}_{\mathrm{unsafe}}|\boldsymbol{s},\boldsymbol{c}) > \epsilon) \le \frac{\mathbb{E}_{\boldsymbol{c}}\left[\ p_\phi(\boldsymbol{a}_{\mathrm{unsafe}}|\boldsymbol{s},\boldsymbol{c})\right]}{\epsilon}, \qquad (8)$$

such that the following objective function is a lower bound of that in Equation 5:

$$\max_{\rho,\phi,\lambda'} \mathbb{E}_{\boldsymbol{c}\sim q_\rho(\boldsymbol{c}|\boldsymbol{s})}\left[\log p_\phi(\boldsymbol{a}|\boldsymbol{s},\boldsymbol{c}) - D_{\mathrm{KL}}(q_\rho(\boldsymbol{c}|\Lambda)||p(\boldsymbol{c})) - \frac{\lambda'}{\epsilon}\mathbb{E}_{\boldsymbol{c}}\left[\ p_\phi(\boldsymbol{a}_{\mathrm{unsafe}}|\boldsymbol{s},\boldsymbol{c})\right]\right]. \qquad (9)$$

For convenience, we write $\frac{\lambda'}{\epsilon}$ as the single hyperparameter $\lambda$ and optimize the log of $p_\phi(\boldsymbol{a}_{\mathrm{unsafe}}|\boldsymbol{s},\boldsymbol{c})$ for better numerical stability. We finally have the lower bound surrogate objective in Equation 6. $\square$

We denote this objective as the *SAFER Contrastive Objective*. Beyond rigorous derivations, this ELBO objective function is intuitively interpretable. The first two terms together act as a contrastive loss that *encourages* safe actions (high likelihood) while *discourages* unsafe ones (low likelihood). Together with the final term, the task variable $\boldsymbol{c}$ is then forced to only contain useful information that does not violate safety. Thus the objective satisfies the earlier goals, allowing for the inference of difficult-to-specify safety constraints through the task variable and discouraging unsafe behaviors. Finally, since SAFER can increase the likelihood of any safe behaviors, the final criteria that the behavioral prior can accelerate downstream policy learning can be met by using safe and successful trajectory data during SAFER training.

**Parametization Choices**  To parameterize the behavioral prior in SAFER, we use the Real NVP conditional normalizing flow, proposed by Dinh et al. (2017), due to it being highly expressive and allowing exact log-likelihood calculations. Next, we parameterize the posterior distribution $q_\rho(c|\Lambda)$ over the safety variable as a diagonal Gaussian. This choice allows computing the KL efficiently, while enabling an expressive task variable latent space. We use a transformer architecture to model the sequential dependency between Gaussian safety variable $\boldsymbol{c}$ and the window of previous states $\Lambda$ (Vaswani et al., 2017). Finally, because the state space is an image pixel space, we also encode each observation to a vector using a CNN. An overview of the architecture is given in Figure 2.

**Training**  It is necessary to use the reparameterization trick to compute gradients across the objective in Equation 6 (Kingma and Welling, 2014). Second, optimizing Equation 6 involves minimizing an unbounded log-likelihood in the second term of the objective. This term can lead to numerical instabilities, if $p_\phi(\boldsymbol{a}_{\mathrm{unsafe}}|\boldsymbol{s},\boldsymbol{c})$ becomes too small. To overcome these issues, we use gradient clipping and freeze this term if it starts to diverge. Psuedo code of the procedure to train SAFER is provided in Appendix D in Algorithm 3 and hyperparameter details are provided in Appendix B. Last, note the objective requires access to unsafe data. We don't advise running an unsafe policy in the real

---

[2]$\lambda'$ can be optimized via gradient descent (Chow et al., 2017) or treated as a hyper-parameter.

---

**Algorithm 1** Accelerating Safe Reinforcement Learning with SAFER

---

**Require:** SAFER Prior $f_\phi$, Safety Posterior $q_\rho(\boldsymbol{c}|\Lambda)$, Safety bound $\eta$, Task $\mathcal{T}$, Window $\Lambda = \{\}$
  **for** step $k = 1, ..., K$ **do**
    $\boldsymbol{s}_k \leftarrow$ current state
    $\boldsymbol{c}_k \leftarrow \mathbb{E}_{q_\rho(\cdot|\Lambda_k)}[\boldsymbol{c}]$               ▷ Compute mean of safety variable
    $\boldsymbol{z}_k \sim \pi_\theta(\boldsymbol{z}|\boldsymbol{s}_k)$               ▷ Sample abstract action from policy
    $\boldsymbol{a}_k \leftarrow f_\phi(\boldsymbol{z}_k; \boldsymbol{s}_k; \boldsymbol{c}_k)$               ▷ Compute action using SAFER
    $s_{k+1}, r_k, \omega_k \leftarrow$ Perform $\boldsymbol{a}_k$ in task $\mathcal{T}$
    Update $\pi_\theta(\boldsymbol{z}|\boldsymbol{s})$ using $(\boldsymbol{s}_k, \boldsymbol{z}_k, \boldsymbol{s}_{k+1}, r_k)$ to maximize $\mathcal{R}_\mu(\boldsymbol{s}_0)$
    Update $\Lambda$ with $\boldsymbol{s}_k$               ▷ Update $\Lambda$ in FIFO order
  **end for**
**Return:** Policy $\pi_\theta(\boldsymbol{z}|\boldsymbol{s})$ for task $\mathcal{T}$

---

world to collect unsafe data. Rather, our method should be applied in a domain where unsafe data already exists or where unsafe data can be collected in simulation (Srinivasan et al., 2020).

### 3.3 ACCELERATING SAFE REINFORCEMENT LEARNING WITH SAFER

When using SAFER on a safe RL task, the goal is to accelerate safe learning by leveraging the behavioral prior in the hierarchical policy $\mu_\psi = \int_{\boldsymbol{z}} f_\phi(\boldsymbol{z}; \boldsymbol{s}; \boldsymbol{c}) d\pi_\theta(\boldsymbol{z}|\boldsymbol{s})$ where the policy parameters of the behavioral prior $\phi$ are fixed and the parameters $\theta$ need to be optimized (Psuedo code of the procedure is provided in Algorithm 1). The policy $\pi_\theta(\boldsymbol{z}|\boldsymbol{s})$ can be learned by any standard RL methods (e.g., SAC (Haarnoja et al., 2018)) that produces continuous actions. To leverage SAFER at inference time for timestep $t$, the current RL policy takes an action in the abstract action space $\boldsymbol{z}_t \sim \pi_\theta(\boldsymbol{z}|\boldsymbol{s} = \boldsymbol{s}_t)$. Using the sliding window of states $\Lambda$, the safety variable posterior computes the distribution over the safety variable

---

**Algorithm 2** SAFER Safety Assurances

---

**Require:** Bound $\eta$, SAFER prior $f_\phi$, safe dataset $\mathcal{D}_{\text{safe}}$, unsafe dataset $\mathcal{D}_{\text{unsafe}}$
  bound $\leftarrow (-\eta, \eta)$
  **define** get_total($\mathcal{D}$):
    total $\leftarrow 0$
    **for** $(\boldsymbol{s}, \boldsymbol{a}, \Lambda)$ in $\mathcal{D}$ **do**
      $\boldsymbol{c} \leftarrow \mathbb{E}_{q_\rho(\cdot|\Lambda_k)}[\boldsymbol{c}], \boldsymbol{z} \leftarrow f_\phi^{-1}(\boldsymbol{a}; \boldsymbol{s}; \boldsymbol{c})$
      **if** $\boldsymbol{z}$ within bound **then**
        total += 1
      **end if**
    **end for**
    **return** total
  **return** $\dfrac{\text{get\_total}(\mathcal{D}_{\text{unsafe}})}{\text{get\_total}(\mathcal{D}_{\text{unsafe}}) + \text{get\_total}(\mathcal{D}_{\text{safe}})}$

---

$c_t$.[3] Because a single safety variable value $c_t$ is required, we fix it at its mean, $\mathbb{E}[\boldsymbol{c}_t] = \int \boldsymbol{c} \, dq_\rho(\boldsymbol{c}|\Lambda_t)$. Finally, SAFER computes the action $\boldsymbol{a}_t = f_\phi(\boldsymbol{z}_t; \boldsymbol{s}_t; \mathbb{E}[\boldsymbol{c}_t])$, the action is taken the environment, and the reward $r(\boldsymbol{s}_t, \boldsymbol{a}_t)$ and safety violations $\omega(\boldsymbol{s}_t, \boldsymbol{a}_t)$ are returned. The action $\boldsymbol{z}_t$ and reward $r_t$ are added to the replay buffer for subsequent RL training.

### 3.4 SAFETY ASSURANCES

To provide concrete and tunable safety assurances under SAFER, for any given bound in the abstract space $\mathcal{Z}$ we develop a technique that estimates the corresponding safety threshold $\epsilon$, where at most $1 - \epsilon$ portion of all actions in data is unsafe (psuedo-code provided in Algorithm 2). Notice that actions that are more likely to be safe and successful are closer to the mean of $\mathcal{Z}$. Therefore, we construct the bound around the mean of $\mathcal{Z}$ using an offline data set of safe and unsafe $(\boldsymbol{s}, \boldsymbol{a})$ pairs. In particular, by fixing any arbitrary range $(-\eta, \eta)$ on each component of the $\mathcal{Z}$ space we compute the corresponding percentage of unsafe actions within the dataset. We take this value as the upper bound of $1 - \epsilon$, the portion of unsafe actions under SAFER. Since the normalizing flow in SAFER is invertible, it allows computing the values in $\mathcal{Z}$ of every $(\boldsymbol{s}, \boldsymbol{a})$ pair, and thus calculating the upper bound is straightforward. Suppose the offline dataset contains sufficiently diverse state-action data that covers most situations encountered by SAFER. Then we would expect the above safety threshold to be rather generalizable and statistically significant (Kääriäinen and Langford, 2005).

## 4 RELATED WORK

**Safe Exploration** A number of related works focus on safe exploration in RL when there is access to known constraint function (Wachi and Sui, 2020; Achiam et al., 2017; Dalal et al., 2018;

---

[3]If there are insufficient states to compute a task window of size $w$ (e.g., at the beginning of the rollout), we pad the available states with 0's in order to construct a window of $w$ states.

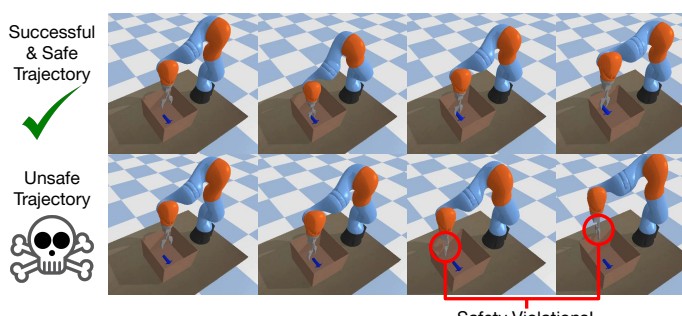

Figure 3: **An example of a task** where the robot successfully and safely grasping an object (top row). Here, the robot reaches into the container and extracts the object *without* touching the container. On the bottom row, the robot performs the same task but commits safety violations by touching the container.

Bharadhwaj et al., 2021; Narasimhan, 2020; Yang et al., 2021; Chow et al., 2018a;b; Achiam and Amodei, 2019; Berkenkamp et al., 2017; El Chamie et al., 2016; Turchetta et al., 2020). In our work, we focus on the setting where the constraint function cannot be easily specified and must be inferred from data, which is critical for scaling safe RL methods to the real world. To this end, a few works consider a similiar setting where the constraints must be inferred from data (Yang et al., 2017; Thananjeyan et al., 2021a). However, these works mainly focused on constrainted exploration and do not consider accelerating learning through learning useful behaviors within the space of safe $(s, a)$ pairs, as is possible with SAFER.

**Demonstrations for safe RL**   Using demonstrations to enable safe RL has received interest in the literature (Rosolia and Borrelli, 2018; Thananjeyan et al., 2020; 2021b; Driessens and Dzeroski, 2004; Smart and Kaelbling, 2000; Srinivasan et al., 2020). Though these works leverage demonstrations to improve safety, they do not handle data from a variety of tasks and enable the acquisition of diverse sets of skills.

**Skill Discovery**   Various works consider learning skills in an online fashion (Eysenbach et al., 2019; Nachum et al., 2019; Sharma et al., 2020; Xie et al., 2021; Konidaris and Barto, 2009). These works learn skills that are used for planning (Sharma et al., 2020) or online RL (Eysenbach et al., 2019; Nachum et al., 2019). In contrast, we focus on a setting with access to an offline dataset, from which the primitives are learned. Further works use offline datasets to extract skills, and transfer these to downstream learning (Pertsch et al., 2020; 2021; Ajay et al., 2021). However, these methods do not consider safety.

**Hierarchical RL**   Numerous works have found it beneficial for policy learning to learn high level primitives using auxiliary models and control these with RL (Singh et al., 2021; Peng et al., 2019; Chandak et al., 2019; Nachum et al., 2018; Hausman et al., 2017; Florensa et al., 2017; Fox et al., 2017a; Dietterich, 1998; Rakelly et al., 2019). Though these works propose methods that are capable of accelerating the acquisition of successful policies, they do not specifically consider accelerating learning with safety constraints and, in some cases, could be susceptible to the issues in Section 2. SAFER considers building a hierarchical policy in the form a behavioral prior that handles safety through encouraging safe actions while discouraging unsafe ones.

## 5   EXPERIMENTS

We evaluate the calibration of the safety assurances introduced in Section 3.4 and how well SAFER encourages both safe and successful policy learning compared to baselines.

### 5.1   EXPERIMENTS SETUP

Recall that SAFER improves data efficiency by first learning a behavioral prior from offline data. To evaluate SAFER, we introduce a suite of safety-critical robotic grasping tasks that are inspired by the game Operation[4].

**Safety-critical Robotic Grasping Tasks**   Inspired by the game Operation, whose primary goal is to extract objects from different containers without touching the container, we construct a set of 40 different grasping tasks, each consisting of a container and object defined in PyBullet (Coumans and Bai, 2016–2021). We collect data from all these tasks to train SAFER but choose 6 of the more complex tasks for evaluation. In each of these tasks, the objects are randomly selected from ones available in PyBullet package, and the containers are generated to fit each object, whose dimensions

---

[4]https://en.wikipedia.org/wiki/OperationGame

Table 1: **Training RL with SAFER**, we give the mean $\pm$ SD success rate and cumulative safety violations across different tasks and initializations. SAFER produces the lowest cumulative safety violations throughout training and outperforms the baseline methods in terms of success rate. Methods without the use of a behavioral prior, namely SAC, are not able to learn during training (of $50,000$ steps). These results demonstrate SAFER is highly effective at encouraging safe learning.

| | Success Rate (%) | | | | | |
| --- | --- | --- | --- | --- | --- | --- |
| | **Task 1** | **Task 2** | **Task 3** | **Task 4** | **Task 5** | **Task 6** |
| SAC | $0.0 \pm 0.0$ | $0.0 \pm 0.0$ | $0.0 \pm 0.0$ | $0.0 \pm 0.0$ | $0.0 \pm 0.0$ | $2.3 \pm 0.0$ |
| PARROT | $0.0 \pm 0.0$ | $12.8 \pm 0.2$ | $25.7 \pm 0.2$ | $16.1 \pm 0.2$ | $33.9 \pm 0.3$ | $6.3 \pm 0.1$ |
| Context PAR. | $5.0 \pm 0.0$ | $24.2 \pm 0.2$ | $27.0 \pm 0.3$ | $0.7 \pm 0.0$ | $7.3 \pm 0.1$ | $12.0 \pm 0.2$ |
| Prior Explore | $1.8 \pm 0.0$ | $1.5 \pm 0.0$ | $3.0 \pm 0.0$ | $1.8 \pm 0.0$ | $1.1 \pm 0.0$ | $1.0 \pm 0.0$ |
| SAFER | $\mathbf{21.0 \pm 0.1}$ | $\mathbf{87.4 \pm 0.2}$ | $\mathbf{89.3 \pm 0.0}$ | $\mathbf{28.1 \pm 0.2}$ | $\mathbf{54.4 \pm 0.1}$ | $\mathbf{83.3 \pm 0.0}$ |

| | Total Number of Safety Violations (Out of 50,000 Steps) | | | | | |
| --- | --- | --- | --- | --- | --- | --- |
| | **Task 1** | **Task 2** | **Task 3** | **Task 4** | **Task 5** | **Task 6** |
| SAC | $2045 \pm 236$ | $876 \pm 117$ | $1055 \pm 216$ | $2736 \pm 147$ | $2188 \pm 405$ | $756 \pm 293$ |
| PARROT | $6332 \pm 3026$ | $307 \pm 291$ | $13 \pm 21$ | $541 \pm 461$ | $2414 \pm 314$ | $932 \pm 844$ |
| Context PAR. | $5929 \pm 2964$ | $1576 \pm 1208$ | $1039 \pm 777$ | $5056 \pm 1778$ | $2796 \pm 624$ | $2085 \pm 1951$ |
| Prior Explore | $6203 \pm 551$ | $2240 \pm 634$ | $2867 \pm 853$ | $4525 \pm 826$ | $4669 \pm 542$ | $2596 \pm 703$ |
| SAFER | $\mathbf{610 \pm 184}$ | $\mathbf{51 \pm 61}$ | $\mathbf{10 \pm 14}$ | $\mathbf{455 \pm 470}$ | $\mathbf{1707 \pm 292}$ | $\mathbf{7 \pm 9}$ |

(heights and widths) are generated randomly. Our agent controls the 5DoF robotic arm and gripper, for which a positive reward ($r(s,a) = 1$) is received when an object is extracted from the box, while a negative reward ($r(s,a) = -1$) is incurred at every time step before the task is complete. The agent incurs a safety violation ($\omega(s,a) = 1$) if the robotic arm touches the box (examples of safe/unsafe trajectories given in Figure 3, examples of the tasks provided in Appendix Figure 10). The states are $48 \times 48$ pixel image observations of the scene collected from a fixed camera.

**Offline Data Collection** To generate the offline data for the SAFER training algorithm, for each robot grasping task we use the scripted policy from Singh et al. (2021) to collect trajectories with a total of $1,000,000$ steps. The scripted policy controls the robotic arm to grasp the object generally by minimizing the absolute distance between objects and the robot. To obtain more diverse/exploratory trajectories, it adds random actuation noise. After collecting the trajectories, for each state-action pair $(s, a)$ in the dataset we provide labels for **i)** safety violation $\omega(s,a) \in \{0,1\}$, and **ii)** whether the pair $(s, a)$ is part of a successful rollout (i..e, $(s, a)$ such that $\mathbb{E}[r(s_T, a_T)|\mu_{\text{data}}, s_0 = s, a_0 = a] = 1$, where $T$ is the trajectory length random variable). To create the state window $\Lambda$ for SAFER training, for each $(s, a)$ in the data buffer we save the previous $w$ states. One can utilize these labels to categorize safe versus unsafe data to train SAFER.

**Baseline Comparisons** We compare against baselines that leverage demonstrations to accelerate learning, including RL from scratch using SAC, PARROT (Singh et al., 2021), a contextual version of PARROT (Context. PAR) that uses a latent variable to help accelerate learning, and Prior Explore, a method that samples from SAFER to help with data collection during training. Because in our problem setting, we infer safety constraints from data and do not have access to such constraints during training, we do not compare against safe RL works that make this assumption. Full details surrounding the baselines are given in Appendix C.

## 5.2 RESULTS DISCUSSION

**Effectiveness of RL training with SAFER** In Table 1 we compare SAFER with the baseline methods both in terms of cumulative safety violations and success rate. We choose a SAFER policy primitive with a safety assurance upper bound that guarantees at most $15\%$ unsafe actions, which empirically maintains a good balance between success and safety. For each downstream task, we then train the RL agent $\pi_\theta$ with SAC for only $50,000$ steps because we are more interested to evaluate the power of the behavioral prior. Overall, we see that SAFER has the lowest cumulative safety violations, indicating that it is the most effective method in promoting safe policy learning. Interestingly, SAFER also consistently outperforms other methods in encouraging successful learning. The strong success rate results of SAFER could be due to the fact that discouraging unsafe behaviors may help refine the space of useful behaviors, better enabling the learning of successful policies.

**Safety Assurance Calibration** We evaluate whether the safe abstract action bound of SAFER computed in Section 3.4 is well calibrated, i.e., the empirical percent of unsafe actions should be

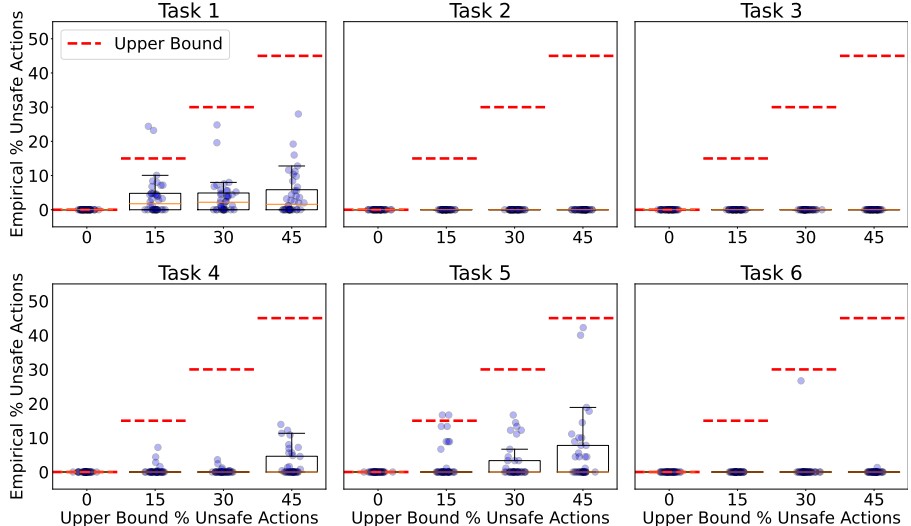

Figure 4: **Assessing the calibration of the SAFER safety assurances** by randomly sampling actions from the prior with various safety upper bounds across different evaluation tasks. Each dot corresponds to the empirical percent of unsafe $(s, a)$ pairs from a single rollout on the task. Overall, we see that the SAFER safety assurances are quite well calibrated.

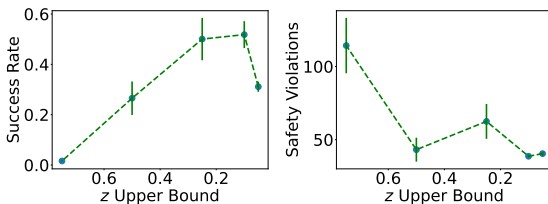

Figure 5: **Assessing the tradeoff between success and safety** varying the safety assurances bound on the abstract action space $\mathcal{Z}$, (referred to as $\eta$ in Algorithm 2). There is an sweet spot where success rate is high and safety violations is low.

less than the upper bound. To study this, we compute the $\mathcal{Z}$-action bound $(-\eta, \eta)$ corresponding to an upper bound of $0\%$, $15\%$, $30\%$ and $45\%$ unsafe actions for SAFER. We compute the percent of unsafe actions by randomly sampling actions from SAFER on each of the evaluation tasks and report the results in Figure 4, which shows that SAFER bounds are indeed well calibrated.

**Success & Safety Tradeoff** In Figure 5 we assess the tradeoff between success and safety by varying the $\mathcal{Z}$-action bound in Algorithm 2. We sweep over different bounds and compute both the success rate and safety violations at the end of training for Task 5. We see that there is a sweet spot with high success rate and low safety violations when the safety assurances bound is close to $15\%$. Interestingly when the bound is too tight (corresponding small $z$ values), both the safety violation and success rate become low, indicating SAFER cannot solve the task without sufficient exploration.

**Impact of latent safety variable** We train SAFER on Tasks 2 and 5 using the contrastive objective in Equation 6 but without the safety variable. In this case, the success rate never exceeds $10\%$ and the safety violations are quite high (see Appendix A for the results for Task 2). In comparisons, with the safety variable the SAFER method has a success rate of at least $60\%$ on both tasks (see Table 1 for details). This suggests that the latent safety variable is crucial for success and safety.

## 6 CONCLUSION

In this paper, we introduced SAFER, a behavioral prior that improves the data efficiency of safe RL when there is access to both safe and unsafe data examples. This is particularly important because most existing safe RL algorithms are very data hungry. We proposed a set of complex safety-critical robotic grasping tasks to evaluate SAFER, investigated limitations of state-of-the-art behavioral priors in safety settings, and demonstrated that SAFER achieves better success rates and enforces safety, and has the following assurance: at state $s$, at most $(1 - \epsilon)\%$ of actions will be unsafe. Future work includes extending the behavioral priors to enforce safety on cumulative or worst case constraints, and applying SAFER to larger-scale problems.

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
