# OpenReview forum: "SAFER: Data-Efficient and Safe Reinforcement Learning Through Skill Acquisition"
_ICLR.cc/2022/Conference — ICLR 2022 Submitted_

### Official Review · Reviewer_LYgh · 2021-10-30

**Correctness:** 3
**Technical Novelty And Significance:** 3
**Empirical Novelty And Significance:** 4
**Recommendation:** 8
**Confidence:** 4

**Main Review:**

This paper is a clear extension of behavior priors to safety domains. The paper argues that the typical BP setup is not adequate for safety contexts because of how they are typically trained vs. out of distribution samples. One question I have here is about the relation between the quality of the dataset (safety of the demonstrated behaviors, and completeness of the demonstrations wrt the full behavioral repertoire) and how large the problem of OOP actually is in this context. I wish more discussion would appear in the paper about this. Conceptually, one could think of creating a better dataset such that all OOP actions would be really outside the "safe" distribution in the dataset. Here the authors choose for a more explicit solution of labeling (automatically) safe and unsafe behaviors and predicting a separate safety variable as a useful signal to learn the difference between safe and unsafe behaviors. I think it makes a lot of sense, although I am not sure a single binary (or graded, but in the end only the mean of a distribution is chosen) will be sufficient for all safety scenarios. Some discussion on the limits of this approach would be appreciated. I am not knowledgeable enough about the specifics of the learning approach (and the contrastive way of optimizing the desired loss) whether there is a contribution here, or whether this is the "smartest" way of doing it. Nevertheless, it seems this is the first time this (BP + safety) is considered so any solution would be sufficient.

I have another issue with the paper when it comes to the "abstract" action space in the function composition of the policy and the BP. I think some of the terminology is confusing a bit, and maybe too close to the deep learning formulation. What I mean by this, is that abstractions in the action space are used very often in MDP contexts, but typically such abstractions are given/modeled/learned (e.g. an abstraction putON(X,Y) abstracting real actions like putON(a,b), putON(b,c), putON(d,a) etc.) or they are part of a behavioral hierarchy (such as in hierarchical RL, with move-to-roomX, with low-level actions in the MDP such as north, south, west, etc). The paper mentions terms like "downstream task", and hierarchical, and abstract, but the definition of Z wrt the MDP (or POMDP for that matter) is not made expliciet (enough). Also, it is unclear how the paper (and BPs in general) relate to many other works in continuous spaces where abstract actions are used which have to be "instantiated" or "grounded" when actually used (e.g. to choose parameters and specific values). To make the paper better, a better description of BPs and how they relate to the problem ((PO)MDP) would be needed, and only after that one should move to a deep learning solutions where Z happens to be positioned in a (relatively random) latent space. This would also make things more clear in terms of learning high-level behavioral choices (Z) and "mappings" from Z to actual actions in the underlying problem space. I checked Singh et al 2021 and the description of BPs there is longer and better, but also lacks a better grounding of concepts in the underlying problem. To make the paper better, and to connect better to related literature, these things need to be described better and more in full.

Overall, a well-done paper with a clear contribution and several good experiments showing the benefits of the approach. The description of the approach can be improved (see issues above, and below, related work, overall story, bibliography, typos, etc).

- "inspired by the game Operation"; I was disappointed it was not about the real game itself :)
- p1:par2:line3 "use a generic offline datasets" (grammar)
- (see also above): explain the non-standard "downstream"
- Figure 2 (and especially the caption text) is not useful at all for me.
- The safety variable can be motivated better
- title section 3.2: "learning THE safety variable"?
- Maybe the chance constrained optimization needs more explanation (technical, especially the relaxation)
- I like the approach in section 3.4. although it also sounds like a practical hack (no offense)
- The related work is ok, but could say more about the safety in RL literature (beyond safe exploration)
- sec5.1:par2:line1 "whose" -> "which"(?)
- sec5.1:par2:line3 "the PyBullet" (grammar, and sentence ends strangely)
- Experiment:table1 I think it is strange that all the baselines perform so badly on the task (success rate). This table says that all methods are not capable at all, and only SAFER works (but only sometimes). This sounds weak. Can you elaborate?
- Experiment:table1 The results for safety violations show large variations over methods and tasks. Is there any pattern here (other than SAFER is more robust)?
- I like the experiments in Fig 5, since they show how safety can hinder raw performance in various ways. Maybe this could be discussed more in full, and earlier in the paper. I think it is relevant.
- Conclusions/future work are quite short, and the bibliography needs polishing.


**Summary Of The Paper:**

This paper builds on previous ideas of behavioral priors in reinforcement learning (RL), more in particular the paper by Singh et al 2021. The main idea is to have a behavioral prior BP function Z x S -> A and a policy over an abstract space Z such that the composition of both will deliver actions that are somehow constrained by the likelihood of actions occurring in a dataset. This dataset is used to obtain the BP in an offline learning sessions, after which standard RL algorithms can learn a policy over Z. Overall this speeds up RL. In this paper this approach is extended with methods that learn the BP in the context of safety conditions (by learning and using an extra safety indicator variable), such that the overall behavior is learned more quickly again but also makes better distinction between safe and unsafe action choices. Much of the paper is about the learning setting (and loss function) to accomplish this. The paper also shows a mechanism to balance performance and safety violations. The approach is tested on a continuous/robotic task of manipulating objects and shown effective, also against some baselines.

**Summary Of The Review:**

Overall, a well-done paper with a clear contribution and several good experiments showing the benefits of the approach. The description of the approach can be improved (see issues above, and below, related work, overall story, bibliography, typos, etc).

---

> ### Author Response · Authors · 2021-11-15
> **Thank you for reading our paper closely and for the thoughtful review! (1/1)**
>
> We thank the reviewer for their positive review. We appreciate the positive comments surrounding the clearness of the contribution, quality of the experiments, and effectiveness of the approach. We respond to the reviewer's comments.
>
> ### Binary Safety Annotations
>
> The reviewer makes a good point that it might be more desirable to have a more expressive notion of safety beyond the binary annotation used in the paper in some settings. We mainly focus on a binary notion of safety violation because it is simpler to define and annotate in many settings than to grade safety continuously. For example, in many settings, it is much easier to fix a binary notion of safety by defining unsafe events (i.e., hitting a curb in autonomous driving) versus developing a continuous notion of safety, which may depend on a complex set of factors (i.e., closeness to the curb, acceleration, etc.). That said, extensions to continuous notions of safety would be exciting directions of future work, because it may allow more expressiveness surrounding different types of safety (likely through modifications to Equation 9 in our paper). For example, it may be possible to label hitting a curb as only somewhat unsafe for autonomous driving and catastrophic accidents (hitting another car) as very unsafe, and in this way, better prevent catastrophic accidents by treating these more harshly.
>
> ### Abstract Action Space and POMDP
>
> We appreciate the comment that we could be clearer surrounding the function of the abstract action space in the safety MDP problem. To address this, we provide an additional description in the paper (Section 2) to relate the abstract action space to the safety mdp problem. In particular, we describe how $z$ serves as a simplified action space in the MDP, where $z$ consists of only actions that assure safety and maximize rewards.
>
> ### Additional Comments
>
> We respond to the reviewer's additional questions here. We revised the grammar and clarity-related comments in the paper, including clarifying downstream, better motivating the safety variable, describing the chance constraint, clarifying the Figure 2 caption, and cleaning up the references.
>
> Table 1: We mainly attribute the performance of the baselines to the difficulty of the tasks (see the SAC baseline) and the improved performance of SAFER due to its capacity to train with much more data (both safe and unsafe examples). Indeed, we performed extensive hyperparameter exploration for the baselines (Appendix C) and still found considerably improved performance under SAFER.
>
> Variations in safety violations: The variations in the safety violations are attributed to the task's difficulty. For example, in task 5, the box is much smaller than in the other tasks, making it much more challenging to solve the task safely.
>
> We thank the reviewer again for their time and valuable feedback.

---

> > ### Comment · Reviewer_LYgh · 2021-11-15
> > **one remark already**
> >
> > Thanks to the authors for clarifications which I will read shortly. Just one remark about the abstract action space already. The "clarification" here is the often-used "Thank you reviewer for your question, and refer to something the reviewer had already read but had a question about" which is not really helpful. Can you be more elaborate here?

---

> > > ### Author Response · Authors · 2021-11-15
> > > **Response**
> > >
> > > We sincerely apologize to the reviewer for lacking description in the comment. Our intentions were to point the reviewer to the new text added to the paper. We add discussion surrounding the abstract action space below.
> > >
> > > For difficult-to-solve safety MDP problems with high dimensional action spaces, the behavioral prior provides a simplification of the action space to only those that are likely safe + successful. For example, in household robotics, a behavioral prior may limit the action space to only those that perform chores while avoiding humans or animals. In this way, the behavioral prior allows exploration of different policies that will be both safe and useful for learning how to do household chores.
> > >
> > > In particular, the safety MDP problem is to solve the constrained optimization problem:
> > >
> > > $ \mu^*\in \textrm{argmax}_{\mu \in \Delta} \, \mathcal R_\mu(s_0): \mathcal W_\mu(s)\leq \epsilon,\forall s\in\mathcal S $
> > >
> > >  where $\mathcal{R}$ is the expected return, $\mathcal{W}$ is the expected safety violation, and $\Delta=\{\mu:\mathcal S\times \mathcal A\rightarrow [0,1], \sum_{a}\mu(a|s)=1\}$ is the set of Markovian stationary policies. Because we are concerned with safety MDPs with complex action spaces that can be difficult to solve with RL-from-scratch (see section 5 where we establish this is the case), we introduce the behavioral prior $f_\phi$ which provides a simplified action space or “abstract action space” $\mathcal{Z}$. The behavioral prior expresses the prior belief that certain actions are more likely to be safe and successful and relates abstract actions $z \in \mathcal{Z}$ to actions $a \in \mathcal{A}$ that are likely to be safe and successful through the following construction $f_{\phi} : \mathcal{Z} \times \mathcal{S} \rightarrow \mathcal{A}$. In terms of the safety MDP, we can think of policies $\pi_\theta$  under fixed the behavioral prior and abstract action space $\mu_\psi=f_\phi ( z; s )$ and $z \sim  \pi_\theta ( z | s)$ as those that are likely to safely solve the task, i.e., $\mathbb{P}(\mathcal{R}_{\mu_\psi} (s_0) \geq r', \mathcal{W} _{\mu_\psi} \leq \epsilon' ) > \xi$ for some threshold $\xi$. Note that there are *many ways* to plausibly express the construction between abstract actions $f_\phi$ that have been considered in the literature (see Skill Discovery and Hierarchical RL sections in the Related Work Section 4), including both designing abstract actions, learning them, or some mix of both.
> > >
> > > The proposed construction is related to methods in the literature that use abstract actions to reduce the space of policies. Our work is most related to methods that do this in an unsupervised way, by learning to extract the relation between abstract actions $\mathcal{Z}$ and useful actions $\mathcal{A}$ through offline data (Pertsch et al., 2020; 2021; Ajay et al., 2021--full citations in the paper). Methods that prescribe semantically meaningful abstract actions (i.e., move to room A or pickup object X in the household robotics example) are closely connected to our construction as well, with the critical difference that $f_\phi$ is not learned in a fully unsupervised manner (Dietterich, 1998--full citation in paper). Further, methods that learning the mapping in an online fashion are related as well (Eysenbach et al., 2019; Nachum et al., 2019; Sharma et al., 2020; Xie et al., 2021; Konidaris and Barto, 2009--full citations in paper). However, the key difference is that we learn the mapping in a fully offline manner. In this work, we focus on extracting the mapping from the abstract action space $\mathcal{Z}$ to safe+useful actions $\mathcal{A}$ using a dataset in a fully unsupervised fashion using a deep learning approach. A positive for this approach is that it is highly scalable with data and can extract complex and diverse actions through data alone. The data can be aggregated from a broad number of sources in a fully offline way, which demonstrates its scalability. However, this technique makes it difficult to interpret the semantic meaning of the learned behaviors, which may be more obvious with a prescribed approach. Overall though, the proposed method is highly useful at learning behaviors that are both safe and successful, through the rigorous construction of the unsupervised objective.
> > >
> > > We hope this response clarifies the reviewer's questions. We would be happy to answer any more questions or elaborate further.

---

> > > > ### Comment · Reviewer_LYgh · 2021-11-20
> > > > **Thanks**
> > > >
> > > > Thanks for this elaborate text. It makes sense, and while reading, I feel that some of the narrative here could help the text of the paper to explain the aspect of abstract actions better. My point also referred to separating the core model (MDP + abstract action) and the function-based/deep-learning solution sufficiently, and I still think opportunities exist to do this better in the paper.

---

> > > > > ### Author Response · Authors · 2021-11-22
> > > > > **Thank you**
> > > > >
> > > > > Thank you for your close reading of the text. We will incorporate the text into the final version.

---

### Official Review · Reviewer_dr7N · 2021-11-02

**Correctness:** 2
**Technical Novelty And Significance:** 2
**Empirical Novelty And Significance:** 3
**Recommendation:** 3
**Confidence:** 4

**Main Review:**

The main contribution of the paper is a modification of existing latent skill-learning algorithms, in particular PARROT (Singh et al.) to incorporate safe learning at test-time by learning a behavioral prior from safe and unsafe offline data. The approach is intuitive, easy to understand, and performs better than two external baselines - PARROT and SAC in terms of exhibiting safer behavior after training.

There are several crucial issues with the paper, that I've listed below:

1. The main drawback of the approach is that it requires an offline data of *both* safe and unsafe behavior for training the latent skill prior. This is fundamentally limiting and goes against the entire principle of safe RL - collecting such a dataset requires multiple failures and hence requires being *unsafe* multiple times during data collection. If the aim of safety is to prevent the robot + it's environment from unintended consequences, implementing this approach on a real robot would lead to multiple failures during data collection and thus by definition be unsafe.

2. Several prior works (both old and new) on safe RL and skill learning are not cited / discussed. Some relevant papers are listed below [a-e].   The papers [a], [b], [c], and those cited in the paper in section 4 emphasize the non-trivial nature of safe RL by describing how it is difficult to collect data while being safe (i.e. safety during training). This paper completely omits discussion of this point and proposes a safe RL method by collecting an offline dataset while being unsafe.

3. In light of the points in 1 and 2, the results on safety violations in Table 1 need to be updated to account for all the safety violations during offline data collection.

4. The paper does not compare with *any* external safe RL baselines. Why are PARROT and SAC the only external baselines compared against? It is important to compare with safe RL papers in the experiments since the main claim of the paper is number of safety violations. Again, for fairness of comparison, the number of safety violations for SAFER should include the violations during data collection.

5. There are no guarantees of any sort on safe behavior. It is important to provide some theoretical guarantees on safe behavior - either during or after convergence. In the current framework, during training would not make sense as the offline dataset collection necessarily involves multiple failures. So, some guarantees on the behavioral prior would help understand the generalization of the method beyond the environments evaluated in the paper.

6. In equation 9, it is unclear how the objective corresponds to a form of contrastive learning. It will be helpful to clarify what is being contrasted with what and why is that helpful, compared to other possible lower bounds on the ELBO.

7. How is it ensured that the policy conditioned on the skill prior, simply doesn't ignore the prior during learning? Is there any constraint on the optimization process to ensure that this does not happen?


There are interesting ideas in the paper around learning a safe behavior prior for latent skill-conditioned policy learning. However, as I pointed out above, there are fundamental issue and assumptions on offline data collection that require multiple safety violations in the data collection process, thereby defeating the purpose of the proposed safe RL approach.

[a] Srinivasan, K., Eysenbach, B., Ha, S., Tan, J. and Finn, C., 2020. Learning to be safe: Deep rl with a safety critic. arXiv preprint arXiv:2010.14603.

[b] Bharadhwaj, H., Kumar, A., Rhinehart, N., Levine, S., Shkurti, F. and Garg, A., 2020, September. Conservative Safety Critics for Exploration. In International Conference on Learning Representations.

[c] Brunke, L., Greeff, M., Hall, A.W., Yuan, Z., Zhou, S., Panerati, J. and Schoellig, A.P., 2021. Safe learning in robotics: From learning-based control to safe reinforcement learning. arXiv preprint arXiv:2108.06266.

[d] Xie, K., Bharadhwaj, H., Hafner, D., Garg, A. and Shkurti, F., 2020, September. Latent Skill Planning for Exploration and Transfer. In International Conference on Learning Representations.

[e] Konidaris, G. and Barto, A., 2009. Skill discovery in continuous reinforcement learning domains using skill chaining. Advances in neural information processing systems, 22, pp.1015-1023.

**Summary Of The Paper:**

This paper proposes an approach for safe policy learning with a behavioral prior conditioned policy. The approach relies on learning a behavior prior from a dataset of safe and unsafe trajectories with chance-constrained variational inference, and using this prior as a latent skill distribution to condition the policy during reinforcement learning. The proposed approach is evaluated against prior skill learning approaches on simulated robot manipulation tasks.

**Summary Of The Review:**

There are interesting ideas in the paper around learning a safe behavior prior for latent skill-conditioned policy learning. However, as I pointed out above, there are fundamental issue and assumptions on offline data collection that require multiple safety violations in the data collection process, thereby defeating the purpose of the proposed safe RL approach.

---

> ### Author Response · Authors · 2021-11-15
> **Thank you for your comments. We have added clarifications and updated the paper to address your concerns!  (1/3)**
>
> We thank the reviewer for the comments. We really appreciate the positive feedback surrounding the intuitiveness of the approach, presentation in the paper, and effectiveness of the method. Thank you!
>
> If the reviewer feels that we have adequately answered their concerns, we sincerely hope *they will consider increasing their score to "accept."*
>
> We respond to the reviewer's comments/questions as follows.
>
> ### Access to unsafe data
>
> We appreciate the comment concerning the assumptions surrounding the data sets used by our method and, in particular, that we assume access to safe and unsafe data to train SAFER. First, we wish to clarify that it is not the goal of our method to collect unsafe data to train SAFER by intentionally committing unsafe behavior. Instead, we assume access to both safe and unsafe data because such data sets are already available for many complex real-world policy learning problems. To this end, to create the data set used in our experiments, we use a scripted policy to generate both safe and unsafe data in order to simulate having access to such a dataset. Intentionally generating unsafe data is not a component of the proposed method, and our method is capable of handling any data previously collected in an offline manner.
>
> Further, it is a very reasonable assumption to have access to both safe and unsafe data and is consistent with prior work in the literature [1]. This data can be due to previous errors in human demonstration, teloperation, or even simulation. Take autonomous driving, for instance. Through both previously learned policies and human demonstrations, there exists data consisting of both safe and unsafe actions performed while driving cars [1, 7]. Closer to the evaluation performed in the paper, human demonstrations are frequently used for robotics [5]. Because humans often make mistakes, such datasets likely contain examples of unsafe behavior [6].
>
> Critically, our method's goal is to leverage pre-existing unsafe data to improve safety in the future (instead of discarding it as is done with current behavioral priors [2]). While, ideally, there wouldn’t be any unsafe behavior in the first place, the fact of the matter is that for many important problems such data exists, and as we show in the paper, it is paramount not to ignore this data (Section 2), because ignoring unsafe data under current methods can easily lead to additional unsafety.
>
> Though these assumptions hold for many critical policy learning problems, it is a good point that it may not make as much sense to use SAFER without access to unsafe data collected in the past, and we add text (Section 5 and 6) to clarify this point.
>
> ### Additional References
>
> Thank you for the additional references. We include these references in the paper and add further discussion concerning how unsafe data may be collected (Section 5).
>
> ### Safety Violations in Table 1
>
> We appreciate the comment surrounding table 1, and we wish to clarify why we did not include all the safety violations incurred during offline data collection. Foremost, it is our assumption that there exists a dataset consisting of both safe and unsafe demonstrations a priori (see “Access to unsafe data” where we establish why it makes sense to use this assumption). Because of this, the safety violations in the data set used by SAFER *still occur* in the datasets for our baselines. However, current methods (such as [2])  ignore unsuccessful/unsafe data during training because they cannot train with this data. On the other hand, SAFER is capable of utilizing unsafe data to improve performance significantly. Therefore, the baselines have the same number of initial safety violations as SAFER, so the comparison between SAFER and the baselines is fair.

---

> > ### Author Response · Authors · 2021-11-15
> > **Thank you for your comments. We have added clarifications and updated the paper to address your concerns!  (2/3)**
> >
> > ### Comparison to Safe RL Baselines
> >
> > We appreciate the comment that we could be clearer surrounding our choice in baselines. We are mainly interested in the critical problem of improving the safety of behavioral priors and therefore focus our comparison against state-of-the-art behavioral priors.  That said, there are a number of limiting factors that make comparisons to standard safe reinforcement learning methods difficult. For one, we do not assume access to the constraint function when training on the tasks, because we focus on the setting where the constraint function cannot be easily specified and must be inferred from data. This means we cannot compare against methods that assume access to a constraint function when learning the tasks [8-11]. It might make sense to use safe rl methods [8-11] if there is access to a constraint function, but we are solving a different problem. In addition, we focus on the setting of adapting to many different tasks, which is different from the objectives of these papers where the goal is to handle one particular problem. Another problematic factor when comparing against standard safe reinforcement methods is that they often use RL from scratch [1, 3, 4], which cannot solve the high-dimensional, sparse reward tasks that we focus on. We see this because our rl from scratch baseline (SAC) cannot solve any of our tasks (success rate ~0%, this result is also consistent with prior work [2]). It is a good point that we can be clearer about our choices, so we include additional motivation for our baselines in the experiment section.
> >
> > ### Guarantees on Safe Behavior
> >
> > While we agree with the reviewer that it could be useful to provide further theoretical guarantees with the method, in this work we mainly focus on thoroughly evaluating the empirical performance of SAFER across many different tasks and ablations (Section 5 and Appendix) and believe the paper should be evaluated in this light. Through these experiments, we clearly established SAFER’s strong empirical performance.
> >
> > That said, we address a number of theoretical safety considerations with the current work. For one, the objective used to train safer is inspired by the safe MDP formulation and offers a lower bound of the chance constrained ELBO (Section 3.2). We additionally offer assurances on safety in the following form: at most $1-\epsilon$ portion of all actions taken in the abstract action space $z$ correspond to unsafe actions in the environment (Section 3.4). These techniques provide theoretical insights into the safety of SAFER. We revise section 3 through the introduction of Proposition 3.1 to better emphasize the theoretical novelty of the work.
> >
> > Finally, we wish to note to the reviewer that many comparable works do not have guarantees on the safety behavior of the method [14-16]. Similarly, many behavioral priors do not have guarantees on the performance of the method [12, 13].  While our work and these do not offer formal guarantees on the method, they are still highly useful, as demonstrated in the empirical results.
> >
> > ### Contrastive Learning
> >
> > We use the term "contrastive" when describing the objective because optimizing the objective contrasts safe and unsafe (s, a) pairs to learn the safety variable $c$. In particular, the objective encourages $c$ to encode useful safety information by showing positive examples (safe data) and negative examples (unsafe data) to learn this latent variable.
> >
> > ### Ignoring the skill prior
> >
> > When training policies on tasks, it is impossible to ignore the SAFER prior because the policy does not perform control directly in the task action space and instead performs control in a separate action space $z$. Every action taken in $z$ is processed by the SAFER prior and is converted into action in the task action space $a$. Further, we bound allowed actions in $z$ to a fixed range (Section 3.4), thereby forcing the actions to be only those that will satisfy a certain level of safety. In this way, it is impossible to ignore the prior.
> >
> > We thank the reviewer again for their time. We would be happy to answer any further questions or comments. We sincerely ask the reviewer to consider *increasing their score to "accept"* if they feel we have adequately responded to their questions.

---

> > > ### Author Response · Authors · 2021-11-15
> > > **Thank you for your comments. We have added clarifications and updated the paper to address your concerns!  (3/3)**
> > >
> > > [1] Krishnan Srinivasan, Benjamin Eysenbach, Sehoon Ha, Jie Tan, Chelsea Finn. Learning to be Safe: Deep RL with a Safety Critic. 2020.
> > >
> > > [2] Avi Singh, Huihan Liu, Gaoyue Zhou, Albert Yu, Nicholas Rhinehart, Sergey Levine. Parrot: Data-Driven Behavioral Priors for Reinforcement Learning. ICLR 2020.
> > >
> > > [3] Nolan Wagener, Byron Boots, Ching-An Cheng. Safe Reinforcement Learning Using Advantage-Based Intervention. ICML, 2021.
> > >
> > > [4] Akifumi Wachi, Yanan Sui. Safe Reinforcement Learning in Constrained Markov Decision Processes. ICML, 2020.
> > >
> > > [5] Fiorini, P., Prassler, E. Cleaning and Household Robots: A Technology Survey. Autonomous Robots 9, 227–235 (2000). https://doi.org/10.1023/A:1008954632763
> > >
> > > [6] Saptari, Adi & Leau, Jia & Ng, Poh Kiat & Mohamad, Effendi. (2014). Human Error and Production Rate Correlation in Assembly Process of Electronics Goods. 10.13140/2.1.3719.2960.
> > >
> > > [7] https://www.theguardian.com/technology/2016/jan/12/google-self-driving-cars-mistakes-data-reports
> > >
> > > [8] Matteo Turchetta, Andrey Kolobov, Shital Shah, Andreas Krause, and Alekh Agarwal. Safe rein- forcement learning via curriculum induction. In H. Larochelle, M. Ranzato, R. Hadsell, M. F. Bal- can, and H. Lin, editors, Advances in Neural Information Processing Systems, volume 33, pages 12151–12162. Curran Associates, Inc., 2020. URL https://proceedings.neurips. cc/paper/2020/file/8df6a65941e4c9da40a4fb899de65c55-Paper.pdf.
> > >
> > > [9] Mahmoud El Chamie, Yue Yu, and Behc ̧et Ac ̧ıkmes ̧e. Convex synthesis of randomized policies for controlled markov chains with density safety upper bound constraints. In 2016 American Control Conference (ACC), pages 6290–6295, 2016. doi: 10.1109/ACC.2016.7526658.
> > >
> > > [10] Felix Berkenkamp, Matteo Turchetta, Angela Schoellig, and Andreas Krause. Safe model-based reinforcement learning with stability guarantees. In I. Guyon, U. V. Luxburg, S. Bengio, H. Wallach, R. Fergus, S. Vishwanathan, and R. Garnett, ed- itors, Advances in Neural Information Processing Systems, volume 30. Curran Asso- ciates, Inc., 2017. URL https://proceedings.neurips.cc/paper/2017/file/ 766ebcd59621e305170616ba3d3dac32-Paper.pdf.
> > >
> > > [11] Yinlam Chow, Ofir Nachum, Edgar Duenez-Guzman, and Mohammad Ghavamzadeh. A lyapunov- based approach to safe reinforcement learning. arXiv preprint arXiv:1805.07708, 2018.
> > >
> > > [12] Avi Singh, Huihan Liu, Gaoyue Zhou, Albert Yu, Nicholas Rhinehart, Sergey Levine. Parrot: Data-Driven Behavioral Priors for Reinforcement Learning. ICLR 2020.
> > >
> > > [13] Anurag Ajay, Aviral Kumar, Pulkit Agrawal, Sergey Levine, and Ofir Nachum. Opal: Offline primitive discovery for accelerating offline reinforcement learning. ICLR, abs/2010.13611, 2021.
> > >
> > > [14] Brijen Thananjeyan, Ashwin Balakrishna, Suraj Nair, Michael Luo, Krishna Parasuram Srinivasan, Minho Hwang, Joseph E. Gonzalez, Julian Ibarz, Chelsea Finn, and Ken Goldberg. Recovery rl: Safe reinforcement learning with learned recovery zones. IEEE Robotics and Automation Letters, 6:4915–4922, 2021a.
> > >
> > > [15] Tsung-Yen Yang, Michael Hu, Yinlam Chow, Peter J. Ramadge, and Karthik Narasimhan. Safe reinforcement learning with natural language constraints. CoRR, abs/2010.05150, 2020b. URL https://arxiv.org/abs/2010.05150.
> > >
> > > [16] Brijen Thananjeyan, Ashwin Balakrishna, Ugo Rosolia, Felix Li, Rowan McAllister, Joseph Gon- zalez, Sergey Levine, Francesco Borrelli, and Ken Goldberg. Safety augmented value estima- tion from demonstrations (saved): Safe deep model-based rl for sparse cost robotic tasks. IEEE Robotics and Automation Letters, 5:3612–3619, 2020.

---

> > > > ### Comment · Reviewer_dr7N · 2021-11-20
> > > > **Thank you for the detailed response. My main concerns still remain.**
> > > >
> > > > Thank you for the detailed response to my comments and for updating the paper based on the comments from all the reviewers. Unfortunately, my main concerns regarding the availability of both safe and unsafe trajectories in the offline dataset, lack of comparisons with safe RL methods, and no guarantees on safe behavior still remain. The main contribution of the paper is learning of a safe prior on top of an existing method PARROT - I am not convinced that the approach is actually safe (both in terms of the empirical results provided, and because there are no guarantees on safety at all). Safety is a very practical consideration, and having only a mild notion of safety-aware learning is not a helpful extension of a prior approach.
> > > >
> > > > - Regarding "we wish to clarify that it is not the goal of our method to collect unsafe data to train SAFER by intentionally committing unsafe behavior", in my understanding this is exactly what the authors did for collecting data in simulation - explicitly simulate unsafe behaviors. I don't see how this could translate to a real world setting where creating unsafe behaviors is actually unsafe. The whole point of safety is to avoid this during learning and data collection.
> > > >
> > > > - Regarding "it is a very reasonable assumption to have access to both safe and unsafe data" the argument would only be convincing if the authors experiment with actual in-the-wild data of real failures that have occurred in a different setting (examples of self-driving etc. as mentioned in the authors' response). Currently all the evaluations are in controlled simulated settings, where the unsafe data is collected in the exact domain of evaluation.
> > > >
> > > > - Regarding "we cannot compare against methods that assume access to a constraint function when learning the tasks " note that the constraint functions need not be shaped functions. Most of the functions used for example in safety critic papers can be such that they only "detect" failures (akin to sparse rewards). So the value is 1 if a failure occurs and 0 if it doesn't, at a particular time-step. So this comparison doesn't require anything beyond what is already assumed in the proposed approach. Unless an existing dataset of failures is used, fair comparison with such online safe rl papers would require addition of failures in the dataset to the count of total failures during training.
> > > >
> > > > In light of the above concerns that still remain, I unfortunately cannot be recommending accept for the paper. I would suggest the authors to have evaluations in realistic settings where the offline dataset of unsafe behaviors is not collected by them, and come up with more formal guarantees on safe behavior. In my understanding, this would require significant improvements over the current paper.

---

> > > > > ### Author Response · Authors · 2021-11-22
> > > > > **We clarify your remaining concerns**
> > > > >
> > > > > Thank you for your response and for clearly describing where you still have concerns!
> > > > >
> > > > > ### Creating the Evaluation Environment & Access to Safe/Unsafe Data
> > > > >
> > > > > We collect a dataset of safe and unsafe behaviors to create a suitable simulated environment to evaluate the baselines and our method. In real-world settings, we do not advise collecting such a dataset. We advise using the method if such a dataset is already available, and we demonstrate that SAFER is highly effective at reducing unsafety in these settings. For many problems, this is a reasonable assumption [7, 1]. Last, the use of simulated environments, like the one proposed in this paper, for evaluation is consistent with many related works [1-4, 8-15].
> > > > >
> > > > > ### Access to Constraint Function
> > > > >
> > > > > Our setting is where we do not have access to a constraint function when adapting to downstream tasks, and the constraints must be inferred entirely from the available dataset. Though other safety-aware reinforcement learning methods use binary notions of safety violations, methods that assume access to the constraint function are not comparisons for our work.
> > > > >
> > > > > ### Theoretical Guarantees
> > > > >
> > > > > Though our work offers theoretical considerations on safety, such as the objective is a lower bound on the chance-constrained ELBO and safety assurances, our work mostly focuses on evaluating the empirical performance of SAFER. Many other works do not establish formal guarantees on safe behavior, however, they are still highly useful at preventing unsafe behavior, as is the same with our work [14-16].
> > > > >
> > > > > ### Main Contribution of the Work
> > > > >
> > > > > Our work offers numerous technical advancements, insights, and novelties. We demonstrate the surprising result that relying on current behavior priors may actually increase unsafe behavior (Section 1 & 2)! Considering that behavior priors are commonly used to bootstrap the performance of RL algorithms this result is highly impactful for their application in safety-critical settings. We offer a rigorous derivation of the SAFER objective. The SAFER objective is novel, generalizable, and can be extended to other parameterizations of behavioral priors to better enforce safety. Also, related works have not considered modeling safety as a latent variable in behavioral priors. We demonstrate that introducing this latent variable can lead to large improvements in both safety and success. Finally, we present a novel way to set the safety assurances under the prior using offline data. Overall, these represent novel and highly significant contributions.
> > > > >
> > > > > Thank you for your time and consideration of the paper.

---

> > > > > > ### Comment · Reviewer_dr7N · 2021-11-28
> > > > > > **Thanks for the followup response.**
> > > > > >
> > > > > > I appreciate the authors' response. The concerns I pointed out in my review and response still remain, unfortunately.
> > > > > >
> > > > > > - I understand that the referenced prior works evaluate safety in simulation, and the authors advise using the method only in settings where prior data is already available. Note that this is different from the main concerns I pointed out in my review and previous response - the concern is that all experiments in the paper explicitly simulate unsafe behaviors, and do not use existing datasets if unsafe behaviors. Doing the latter in my understanding is not a trivial application of the method because it would require addressing multiple issues of not having data in the exact setting of evaluation, and that of domain shift.
> > > > > >
> > > > > > - I don't follow the argument regarding constraint functions. In my understanding it should be possible to compare against prior works that assume constraint functions, simply by making those constraints binary (this is equivalent to only being able to know when a failure occurs, which is an assumption in the proposed method as well).
> > > > > >
> > > > > >
> > > > > > - By formal safety guarantees I did not mean derivation of the objective, but some formal bound on safe behavior. The authors point out that it is fine to not have this because some other prior works also don't have bounds of this form. Note that this argument doesn't really hold on its own - the other prior papers that do not have theoretical bounds also have different assumptions, problem settings, and experiments in the online setting without explicitly simulating unsafe behavior (I am sorry, but we can't just compare one aspect and ignore the others). In general, I definitely agree that strong empirical demonstration of safe behavior in real-world settings can be a good substitute for formal bounds on safety, but as I pointed out above and in my review, the setting and experiments in the current paper do not reflect real-world considerations.
> > > > > >
> > > > > >
> > > > > > I thank the authors for their time in responding to the reviewers' concerns, and for participating in the discussion. For completeness, I'll summarize by repeating what I mentioned in the previous response. In light of the above concerns that still remain, I unfortunately cannot be recommending accept for the paper. I would suggest the authors to have evaluations in realistic settings where the offline dataset of unsafe behaviors is not collected by them, and come up with more formal guarantees on safe behavior. In my understanding, this would require significant improvements over the current paper.

---

### Official Review · Reviewer_GszC · 2021-11-05

**Correctness:** 3
**Technical Novelty And Significance:** 3
**Empirical Novelty And Significance:** 2
**Recommendation:** 6
**Confidence:** 3

**Main Review:**

> We rewrite the optimization 5 into its equivalent

Regarding the expression "equivalent": Eq. 5 is a mathematical program, while eq 6 is Lagrangian-like term. Eq. 6 is not a mathematical program per-se. You may want to write it as a saddle-point problem if you want to talk about "equivalent optimization", or say something related to the KKT conditions.

> Equation 8 and 9

You use $\theta$ while I think you mean to use $\phi$.

> we propose the specific policy structure µψ = fφ ◦ πθ, where fφ ◦ πθ(a|s) := R z∈Z fφ(z, s)dπθ(z|s), and ψ = (φ, θ)

I think that talking about function composition is appropriate here because the underlying model is inherently probabilistic. All this is saying is that you have a mixture model, which is a clear concept when explained in those terms.

Also, in your math expression, I don't understand why you have "a" on the lhs but not the rhs.

# Actionable feedback

Be clear about the problem setup. "Safety" can mean a lot of things depending on the framework, and the flavor considered in this work is not a usual one of the kind that we have in CMDPs for example. I was personally struggling to grasp the motivation and the real-world grounding of the problem setup. The operation game for the robot is concrete, but arguably a rather artificial safety-critical setup. What would be a real-real world scenario where your framework would make sense (compared to the alternatives)? I suggest you expand on this in the intro.

**Summary Of The Paper:**

This paper is an extension of "PARROT" by Singh et al. (2021), which does a form of hierarchical pretaining from offline datasets to a setting where preferences need to be expressed on the kinds of behaviors that ought to be generalized. The authors use the expression "safety" to refer to what their work tries to accomplish, but I think that a more direct description of their work is in terms of preference specification via binary variables.

My reference point to understand this work is to think about hierarchical reinforcement learning, especially the probabilistic viewpoint laid out by Daniel et al. (2016). In this perspective, we think of the "termination functions" (or initiation) in the options framework in terms of the sequence of Bernoulli variables that they induce. Similarly, the proposed SAFER approach introduces an auxiliary "safety" variable "c" which needs to be inferred at execution time.  In the proposed framework, it is as if we were to learn a mixture model where instead of having a categorical distribution over options (discrete), we would instead be using a continuous (Gaussian) distribution to represent our distribution over latent options. The particularity of this line of work (PARROT) is that this Gaussian latent variable is learned in an unsupervised manner over a dataset of successful solutions across different tasks, rather than a purely downstream-oriented manner.

The new component that this paper adds to PARROT is this additional latent "safety" latent variable. (Note: this paper is not very self-contained: I had to read carefully the PARROT, which I didn't know beforehand, to make sense of the proposal.) One complication that arises from this modification is to now enforce that this new "safety" latent behaves the way that is intended to: that is, to steer/control the induced behavior to act "safely" when told to do so. The solution proposed by the authors is to view this as a chance-constrained optimization problem. This allows them to write the safety satisfaction requirement in probabilistic terms (constraint violation), which they then turn into a constrained program where the probabilistic constraint becomes a deterministic one via Markov's inequality. The authors then *do not solve* the resulting problem as a constrained problem, but rather choose to work with a regularized unconstrained approximation coming from the Lagrangian.

A last important conceptual idea of the paper is that of calibration (which they call "safety assurance") of that safety variable. The problem arises from the fact that in adding this safety knob, we need to ensure that the constraint violation coefficient specified in the chance-constrained relaxation does indeed coincide empirically with the level set by the designer. Here, the authors leverage the properties of normalizing flows to estimate this threshold from historical data.

**Summary Of The Review:**

My main criticism of this paper is that it presents a fairly specific idea, that completely hinges on very recent prior work by Singh et al. 2021: to the point that the submitted paper is undecipherable unless you read Singh et al. (2021) first (which I had to do). That being said, the paper presents an interesting new perspective and builds on chance-constrained optimization and constrained optimization (but the authors choose not to go deeper in this direction, perhaps because of technical difficulties that are not mentioned in the paper).

---

> ### Author Response · Authors · 2021-11-15
> **Thank you for your thoughtful review. We have added clarifications and updated the paper to address the minor concerns: (1/1)**
>
> We thank the reviewer for their detailed reading of the paper and valuable feedback and appreciate the positive comments surrounding the interesting perspective presented in the paper. We respond to the reviewer's comments.
>
> If the reviewer feels we have adequately addressed their comments, we would appreciate it if they *consider increasing their score to "accept."*
>
> ### Problem Setup
>
> We appreciate the comment that we could be more explicit about the problem setup. We clarify the problem setup below and also modify the introduction & background per the reviewer’s feedback to reflect this further motivation and make the paper fully self-contained.
>
> This paper focuses on high-dimensional reinforcement learning settings (i.e., learning from pixels) with sparse rewards and safety considerations that are difficult to specify in a constraint function. Numerous different policy learning problems fit this description. For example, one application is household robotics from image observations with binary success/failure rewards and safety constraints related to whether the robot behaves dangerously in a home. Another example is autonomous driving from high-dimensional observations with binary rewards for whether reaching a destination is accomplished or not and complex safety constraints concerning driving safely. Because RL from scratch is unsuccessful in these high-dimensional sparse reward settings (the success rate stays close to 0% as established in the literature [2] and in our work), one promising solution is to use behavioral priors [1, 2]. Intuitively, if trained on, say, a household robotics task, we would expect these methods to learn safe and useful behaviors (e.g., handling objects carefully or not harming animals/people). However, as we demonstrate in Sections 1 & 2, current methods are actually poorly equipped to do so. Current behavioral priors may lead to unsafe behavior (i.e., breaking things or harming people). As we show, SAFER can overcome these limitations and extract skills that lead to safe and successful behavior, so is the better choice in these settings. For the household robotics example, this means SAFER will learn skills that handle objects with care and avoid animals/people in the environment.
>
> ### Specificity of the Idea
>
> Though the reviewer indicates the idea presented in the paper is specific and concerns recent work, we believe our work makes considerable advances on a promising new direction in reinforcement learning---the behavioral prior. As our work and prior work demonstrates, it is challenging to solve complex, real-world RL problems with current RL techniques [1 , 2]. Because behavioral priors have shown considerable promise and are already commonly used, ensuring that such methods are also safe is critical. Our work demonstrates the crucial insight that behavioral priors may behave unsafely because they ignore unsafe/unsuccessful data, and we introduce SAFER to overcome these issues.
>
> Further, our framework is highly versatile and can be easily extended to many other methods. The derivation and objective provided in section 3.2 provide a general framework to enforce safety in behavioral priors. For instance, under our general framework, it is possible to consider other parameterizations of the behavioral prior and safety variable. Overall, our work represents a significant contribution for these reasons.
>
> ### Additional Comments
>
> We revise the additional comments in the paper. In particular, we apologize for the incorrect usage of the composition. Further, having $a$ on the lhs was a typo and we revised this in the new version.
>
> Again, we appreciate the reviewer's time and valuable comments. If the reviewer feels that we have adequately addressed their concerns, we would sincerely appreciate it if they *consider increasing their score to “accept.”*
>
> [1] Avi Singh, Huihan Liu, Gaoyue Zhou, Albert Yu, Nicholas Rhinehart, Sergey Levine. Parrot: Data-Driven Behavioral Priors for Reinforcement Learning. ICLR 2020.
>
> [2] Anurag Ajay, Aviral Kumar, Pulkit Agrawal, Sergey Levine, and Ofir Nachum. Opal: Offline primitive discovery for accelerating offline reinforcement learning. ICLR, abs/2010.13611, 2021.

---

> ### Comment · Area_Chair_2A7o · 2021-11-28
> **Please participate in discussions!**
>
> Hi,
>
> Please read the authors' response, and engage in discussions with them. Have they adequately addressed your concerns?
>
> Thank you,
> Area Chair

---

### Official Review · Reviewer_31VF · 2021-11-07

**Correctness:** 4
**Technical Novelty And Significance:** 3
**Empirical Novelty And Significance:** 3
**Recommendation:** 5
**Confidence:** 4

**Main Review:**

Strength:
The problem is important and interesting.
The paper was well written and easy to understand.
The experimental results also show the methods have better performance than the benchmarks.

Weakness:
The key components of the methods, including using conditional normalizing flow to represent the behavioral prior, using chance constraints to represent unsafe constraints, using ELBO to optimize the chance constraints are all well-known standard treatments in the field of reinforcement learning. The authors provided heuristics on the choices of these tools but lack rigorous analysis on the proposed methods. Given the authors only one environment to evaluate the proposed method, it is unknown how well the methods could generalize to other problems.

It is suggested that authors add more theoretical analysis to unveil the novelty and insight of the method and more diverse experiments to show the strength and limits of the methods in a more comprehensive way.

**Summary Of The Paper:**

This paper proposes a new safe RL method, SAFER, that uses a behavioral prior learning algorithm to accelerate policy learning under safety constraints. SAFER learns to extract a safety variable from offline data and the safe primitive skills over abstract actions in different scenarios via contrastive training. The authors demonstrate the effectiveness of the methods with several complex safety-critical robotic grasping tasks inspired by game Operation. Results show that SAFER no only outperforms baseline methods in learning successful policies but also enforces safety more effectively.



**Summary Of The Review:**

It is an interesting paper but lacks insight into the novelty and the application scope of the proposed method

---

> ### Author Response · Authors · 2021-11-15
> **Thank you for your comments. We have updated the paper to address your concerns! (1/2)**
>
> We thank the reviewer for their comments. We truly appreciate the positive comments surrounding the *importance of the work, quality of the writing, clearness, and strength of the method. Thank you!* We answer the reviewer's questions and comments below.
>
> If the reviewer feels like we have adequately answered their questions, we *would appreciate it if they consider changing their score to "accept."*
>
>  ### The novelty of the method
>
> While the reviewer points out previous methods use components considered in this work, our work offers numerous novelties. First, *few solutions* exist to address the highly critical problem of policy learning in high dimensional, sparse reward settings where safety constraints must be inferred from data. This problem has applications to many important reinforcement learning areas, such as robotics and autonomous driving. We offer an effective solution to this problem, thereby offering significant novelty in our work, and we sincerely hope the reviewer can take into account this fact.
>
> Further, our work offers novelties through both technical insights and methodological advancements. We analyze current behavioral priors (Sections 1 & 2) and point out their critical and surprising shortcomings and how relying on them may actually increase unsafe behaviors!
>
> Given that behavioral priors are standard tools to bootstrap the performance of reinforcement learning algorithms, this result is highly impactful for their application in safety-critical settings. In addition, we show how we can overcome these issues through the SAFER contrastive objective to infer the safety variable (Section 3). This objective is novel, highly generalizable,
> and can easily be extended to other parameterizations of behavioral priors to better enforce safety. We present a rigorous derivation of this objective by lower bounding the ELBO through the Markov inequality. Moreover, the literature has not considered modeling safety as a latent variable in behavioral priors, and we show that introducing this latent variable can lead to considerable improvements in both safety and success.
>
> In addition, we present a novel and highly useful technique to compute the safety assurances afforded under the prior (namely, at any state $s$ at most $1-\epsilon$ proportion of the actions are unsafe), and we demonstrate it leads to reliable estimations (Section 3.4). Last, we introduce a novel suite of 40 tasks to evaluate complex, safe reinforcement learning methods (Section 5.1). All in all, these contributions represent considerable and highly significant advancements.
>
> ### Evaluation framework
>
> Though the reviewer indicates it could be possible to introduce more diverse experiment settings, we are interested in solving complex reinforcement learning problems in high-dimensions where safety constraints need to be inferred from data (i.e., learning from pixels with high-dimensional continuous actions). These problems are significant because they are similar to real-world situations encountered in robotics, where the goal is to learn from high-dimensional observations over a complex action space and difficult to specify safety constraints, such as autonomous driving. Many current safe reinforcement learning environments do not satisfy these requirements, so we focus on developing and introducing the benchmark in the paper. While the benchmark is situated in one environment, it comprises many different tasks that state-of-the art RL algorithms fail to solve and demonstrates that our method can solve different problems successfully (Section 5.1). Finally, we perform many rigorous ablations (Section 5.2 and Appendix) to demonstrate the method's effectiveness. Overall, we clearly establish the superior performance of SAFER through extensive experimental evaluation.

---

> > ### Author Response · Authors · 2021-11-15
> > **Thank you for your comments. We have updated the paper to address your concerns! (2/2)**
> >
> > ### Theoretical analysis
> >
> > While it could be useful to provide further theoretical guarantees with the method in future work, in this work we mainly focus on thoroughly evaluating the empirical performance of SAFER across many different tasks and ablations (Section 5 and Appendix). Through these experiments, we clearly established SAFER’s strong empirical performance. Note that work in our problem area is quite nascent, with many works providing empirical solutions  [1, 2, 3, 4, 5]. While our work and these do not offer formal guarantees, it is important to provide empirical solutions to demonstrate that the problem can be effectively solved. In the future, it would be very interesting to do further theoretical analysis to understand the effectiveness of the method.
> >
> > That said, we do offer theoretical contributions in the work, and we acknowledge that we could do a better job highlighting these. Therefore, we revise section 3 to better highlight the contributions, including the introduction of Proposition 3.1, to more rigorously describe the lower bound that we introduce on the chance constraint. Further, we additionally offer assurances on safety in the following form: at most $1-\epsilon$ portion of all actions taken in the abstract action space $z$ correspond to unsafe actions in the environment (Section 3.4).
> >
> > We thank the reviewer again for their time and positive comments. We would be happy to answer further questions surrounding the work, and we sincerely ask the reviewer to consider increasing their score to "accept" if they feel we have adequately responded to their concerns.
> >
> > [1] Avi Singh, Huihan Liu, Gaoyue Zhou, Albert Yu, Nicholas Rhinehart, Sergey Levine. Parrot: Data-Driven Behavioral Priors for Reinforcement Learning. ICLR 2020.
> >
> > [2] Anurag Ajay, Aviral Kumar, Pulkit Agrawal, Sergey Levine, and Ofir Nachum. Opal: Offline primitive discovery for accelerating offline reinforcement learning. ICLR, abs/2010.13611, 2021.
> >
> > [3] Brijen Thananjeyan, Ashwin Balakrishna, Suraj Nair, Michael Luo, Krishna Parasuram Srinivasan, Minho Hwang, Joseph E. Gonzalez, Julian Ibarz, Chelsea Finn, and Ken Goldberg. Recovery rl: Safe reinforcement learning with learned recovery zones. IEEE Robotics and Automation Letters, 6:4915–4922, 2021a.
> >
> > [4] Tsung-Yen Yang, Michael Hu, Yinlam Chow, Peter J. Ramadge, and Karthik Narasimhan. Safe reinforcement learning with natural language constraints. CoRR, abs/2010.05150, 2020b. URL https://arxiv.org/abs/2010.05150.
> >
> > [5] Brijen Thananjeyan, Ashwin Balakrishna, Ugo Rosolia, Felix Li, Rowan McAllister, Joseph Gon- zalez, Sergey Levine, Francesco Borrelli, and Ken Goldberg. Safety augmented value estima- tion from demonstrations (saved): Safe deep model-based rl for sparse cost robotic tasks. IEEE Robotics and Automation Letters, 5:3612–3619, 2020.

---

> ### Comment · Area_Chair_2A7o · 2021-11-28
> **Please participate in discussions!**
>
> Hi,
>
> Please read the authors' response, and engage in discussions with them. Have they addressed your concerns?
>
> This is an important step in making sure we evaluate this paper fairly and accurately.
>
> Thank you,
> Area Chair

---

### Comment · Reviewer_LYgh · 2021-11-20
**General evaluation**

Thanks to the authors for providing a lot of additional explanations and modifications. Based on this and the reviews, I am still positive about the paper. ICLR restricts the positive (accept) evaluations to include something equivalent to "just above the threshold" and a firm "accept". I chose the latter, since I think that the first is too low. However, my evaluation should be read as something in between the two. I share the many concerns of the other reviews, for example when it comes to the matter of "having safe/unsafe demonstrations", and the experimental evaluation. Compared to the safety literature, there are some questions to ask as whether the approach is realistic enough on this aspect, for example. Overall though, I do like the approach and the paper, and the authors have provided extensive and relevant reactions to all reviewers.

---

> ### Author Response · Authors · 2021-11-22
> **Thank you**
>
> Thank you for the positive feedback! We sincerely appreciate it.

---

### Comment · Area_Chair_2A7o · 2021-11-28
**Please participate in discussions!**

Dear reviewers,

If you haven't already done so, please read the authors' response, and engage in discussions with them. Please specify whether they have adequately addressed your concerns or not. If they have, please update your score accordingly.

Thank you,
Area Chair

---

### Decision · Program_Chairs · 2022-01-20

**Decision:**

Reject

**Comment:**

The paper provides an algorithmic framework to accelerate RL through Behavioral Priors, while having some notion of safety incorporated. The reviewers are divided about this paper:

On the positive side, some of the reviewers consider the problem important, and the experimental results reasonable and promising.

On the negative side, reviewers raised issues such as
1) The paper is on a heuristic side.
2) No formal guarantee on the safety is provided.
3) The paper is not as self-contained as it should be, as it relies much on Singh et al. (2021).
4) The algorithm requires access to unsafe offline data.

I do not give the same weights to all these concerns. For example, even though (d) is an issue in some applications, it is alright for others. What concern me most are (1) and (2).

A method for safety that is only evaluated empirically and does not have any formal guarantee cannot be used for safety critical tasks. I realize that some other published papers may have the same issue. But given that this is a real concern, and that two out of four reviewers believe that the paper should not be accepted, unfortunately I cannot recommend acceptance of this paper, especially given the competitiveness of this conference.

P.S: I also noticed that in the proof of Proposition 3.1, an expectation term $E[p_\phi(a|s,c)]$ in Eq. (9) is replaced by a $\log p_\phi(a|s,c)$. This requires more justifications.